# Visual Duration but Not Numerosity Is Distorted While Running

**DOI:** 10.3390/brainsci12010081

**Published:** 2022-01-05

**Authors:** Irene Petrizzo, Giovanni Anobile, Eleonora Chelli, Roberto Arrighi, David Charles Burr

**Affiliations:** Department of Neuroscience, Psychology, Pharmacology and Child Health, University of Florence, 50135 Florence, Italy; irene.petrizzo@unifi.it (I.P.); giovanni.anobile@unifi.it (G.A.); eleonora.chelli@stud.unifi.it (E.C.); roberto.arrighi@unifi.it (R.A.)

**Keywords:** time perception, numerosity perception, physical activity, magnitude system, atom

## Abstract

There is increasing evidence that action and perception interact in the processing of magnitudes such as duration and numerosity. Sustained physical exercise (such as running or cycling) increases the apparent duration of visual stimuli presented during the activity. However, the effect of exercise on numerosity perception has not yet been investigated. Here, we asked participants to make either a temporal or a numerical judgment by comparing the duration or numerosity of standard stimuli displayed at rest with those presented while running. The results support previous reports in showing that physical activity significantly expands perceived duration; however, it had no effect on perceived numerosity. Furthermore, the distortions of the perceived durations vanished soon after the running session, making it unlikely that physiological factors such as heart rate underlie the temporal distortion. Taken together, these results suggest a domain-selective influence of the motor system on the perception of time, rather than a general effect on magnitude.

## 1. Introduction

One of the core missions for perceptual systems is to provide the brain with reliable information about the environment to enable efficient interaction with nearby objects via goal-directed actions. Sensory estimates need to be accurate and precise in many dimensions as objects (and events) are internally represented within a multidimensional space encompassing many properties, including spatial position, time of occurrence, and numerosity. As these variables often correlate with each other (for example, it takes more time to walk a longer distance or to pick up more cherries from a table), it has been proposed that there exists within the parietal lobe of the human brain a shared magnitude system to process information about space, time, and quantity via a single mechanism. This idea, known as “A theory of Magnitude-ATOM”, has been extensively tested by measuring the extent to which the processing of information in one of the ATOM dimension distorts estimates in another [1]. For example, several studies have reported that large visual stimuli are perceived to last longer than smaller ones presented for the same physical duration [2]. Duration estimates are also significantly affected by stimuli numerosity: numerous stimuli are perceived to last longer than less numerous ones [3].

A shared mechanism processing quantitative information in multiple dimensions may be beneficial in providing a unique interface between the perceptual and the motor systems, sub-serving the transfer of sensory information between them. In line with this idea, voluntary movements performed during the presentation of visual stimuli can affect perceived duration. For example, at the time of saccades, duration is considerably compressed, by about 50% [4]. Similar (but weaker) compression also occurs when task-unrelated hand movements, either a series of discrete temporal patterns or continuous actions, are made during the presentation of visual stimuli [5,6]. Action does not always cause compression: a saccadic movement to a clock’s second hand can induce the illusion of temporal expansion, sometimes referred to as “saccadic chronostasis” [7].

Recently, a new motor-induced temporal illusion has been reported, where repetitive hand-tapping can distort the perceived duration of stimuli subsequently presented around the tapping area [8]. The perceived duration of visual stimuli is significantly compressed (by about 30%) after fast tapping and expanded after slow tapping. Motor adaptation of this type does not only distort perceived time but also perceived numerosity [9] and spatial distance [10]. However, motor adaptation does not affect the perception of stimulus speed, suggesting that the motor system might selectively interact with the processing of parietal information, without affecting that of earlier levels of visual processing.

A recent study from Lambourne et al. investigated the role of physical exercise on a temporal comparison task and found that visual stimuli presented during sustained aerobic exercise executed with the lower limbs (cycling) induced an expansion of perceived duration of about 15% [11]. However, the study did not address whether these effects were selective only for temporal estimates or whether they generalize to other quantitative dimensions, as would be suggested by the ATOM Theory. A second question is whether sensory distortions are induced only during the execution of actions or whether the distortion persists after completion of motor activity, indicating a relatively long-term recalibration of the sensory system after physical activity. If the effect induced by self-motion were to disappear immediately after the end of the physical activity, it would indicate that the distortion of perceived duration is related to the movement itself and not to the other physiological variables that are changed during the activity and take time to revert to baseline levels. While both hand-tapping and cycling (or running) can be categorized as “self-motion”, they are very different in terms of effectors (upper or lower limbs) and frequency, and while hand-tapping could more closely resemble the action of counting items scattered on a surface, cycling is a non-goal-directed action. If a different influence of action on numerosity depending on the type of motion was to be found, it would suggest a selective interaction between action and numerosity, with types of motion more closely related to counting being more prone to distortion of numerosity.

In our new paradigm, the participants made a temporal or numerosity comparison (separate sessions) in three different conditions: at rest, during sustained physical exercise (running on a treadmill), or immediately after the exercise. In line with previous reports, perceived duration was expanded during the motor routines; however, estimates of numerosity were almost completely unaffected. Furthermore, time judgements were not distorted for stimuli presented soon after the end of the physical activity, although several physiological variables, such as heart rate, remained altered relative to the baseline, suggesting that distortions of time occur only during the execution of actions, with rapid recalibration after the activity is completed.

## 2. Materials and Methods

### 2.1. Participants

A total of fifteen participants took part in the study (8 females, 7 males, mean age = 27.3, SD = 6.4; 11 were naive to the purpose of the study, and 4 were authors). All participants had normal or corrected-to-normal visual acuity and provided written informed consent and a medical certificate for non-competitive physical activity. Each experiment was conducted on a different day, with the order of experiments pseudo-randomized across the participants. The duration of each experimental session was around 2 h per participant. The research was approved by the local ethics committee (“*Commissione per l’Etica della Ricerca*”, University of Florence, 7 July 2020, n. 111).

### 2.2. Apparatus

The experiments were conducted in a dimly lit, low-noise room with participants standing or running on a treadmill (JK Fitness Supercompact 48) at approximately 90 cm from the monitor (Telefunken Smart TV 43″). Heart rate was measured with a Garmin Forerunner 55 smartwatch paired with an HRM-Dual Heartrate strap. The setup allows continuous monitoring of the participants’ heart rate (temporal resolution: 1 Hz) via Bluetooth. Stimuli were generated and presented with PsychToolbox 3 routines [12] in Matlab 2016b (The Mathworks, Inc., Natick, MA, USA).

### 2.3. Duration Perception While Running

The task was similar to that used by Lambourne et al. [11] (Figure 1A). In each trial, the participants were presented with a central visual stimulus (24 cm × 24 cm blue square, approximately 15° at the viewing distance of 90 cm). The participants judged the stimuli as the “same” or “different” compared with a previously memorized stimulus lasting 600 ms (reference). The nine test stimulus durations were logarithmically spaced around the reference: 284, 342, 413, 498, 600, 723, 872, 1052, and 1268 ms.

An experimental block comprised seven steps (Figure 1A). The first was a training session, where the reference stimulus was presented 5 times sequentially with no response required (encoding phase). Then, all the test durations were presented once in random order, and for each of them, the participants reported whether they had the same or a different duration from the reference. During this phase, response feedback was provided (correct or incorrect, signaled by a change in color of a central fixation point). When the participant reached 80% correct responses, the training stopped, otherwise another block of nine trials started. After training, the encoding phase started. In this phase, the participants were first presented with the reference stimulus 5 times (as at the beginning of the training); then, a first decoding phase started (baseline T1) after three minutes of rest from the encoding. The decoding phase consisted of 66 test trials, where each test of a different duration from the standard was presented 6 times, and the standard duration was presented 18 times. In each trial, participants reported whether each stimulus had the same or a different duration relative to the reference.

After the baseline T1, there was a new encoding phase, followed immediately by the running phase, lasting 3 min. During running, the treadmill speed was continuously adjusted by the experimenter to keep the participant’s heart rate around 80% of the maximum heart rate for his/her age, according to the formula: 208 − 0.7*(participant’s age) [13]. At the end of the first three minutes of running, when the target heart rate was reached and maintained, the participants started a second decoding phase (identical to the first) while they kept running. During the running phase, the speed was constantly monitored and regulated to maintain heartbeat as close as possible to the target. The test phase lasted about 5 min, with the total running time of the block lasting 8 min. Once the participants had stopped running, and the heartbeat reverted to the baseline level (±10 bpm), a new encoding phase started, followed by a second baseline (T2) measurement. After a short break (about 10 min), the whole procedure (apart from the training) was repeated in the same temporal order. At the end of the experimental session, each participant had completed 2 blocks per condition, for a total of 132 trials for each block (396 in total).

### 2.4. Duration Perception after Running

As for the previously described experiment, in each trial the participants were asked to judge whether a stimulus had the same or a different duration of the memorized reference (600 ms), with identical stimuli to those described above (Figure 1B). As before, a block started with training followed by a decoding phase, a rest phase (this time lasting 8 min, to set the same interval between the decoding and the encoding phase as in the experimental phase), and a baseline (T1). After the baseline, a new encoding phase was performed before starting the running phase. During the first three minutes of running, the speed was manipulated to make the participant’s heartbeat reach the target value (as in Exp 1). Once the target heartbeat had been reached, the participants kept running for an additional 5 min without being presented with any stimulus. During the running, the speed was adjusted to keep the heartbeat near the target value. After 8 min of activity, the treadmill was stopped, and the participants immediately started the test phase (test after run). In this experiment, we did not test the baseline at T2. After the running phase, the participants were allowed to take a break and rest, and after making sure that the heartbeat had returned to baseline levels (±10 bpm), the whole procedure was repeated, apart from the training.

### 2.5. Numerosity Perception While Running

This experiment was procedurally identical to the measurements of duration during running (see Figure 1A), but in this case, the stimuli were circular arrays of black and white dots presented in the center of the screen. Each array had a diameter of 47 cm (about 29° at the average viewing distance of 90 cm), and each dot had a diameter of 1.5 cm (0.95°). In each trial, a single array was presented for 200 ms (to avoid serial counting). The reference numerosity was 24 dots, while the test numerosities were logarithmically spaced around the standard: 11, 14, 17, 20, 24, 29, 35, 42, or 51 dots.

### 2.6. Running Variables and Heartbeat Parameters

Table 1 reports the descriptive statistics of heart rate, running speed, and total number of steps for each experiment. In the numerosity-while-running experiment, the heartrate of one participant was not collected due to technical failure. The heart rate and running speed were calculated excluding the first 3 min of warm up (the period in which the target heartrate was gradually reached, see Figure 2). The number of steps refers to the whole running period (3 min warm up plus 5 min of running). Baseline heartrates were obtained by averaging all heartrates at resting state across the three experiments.

Figure 2 illustrates the average heart rate across the session, with the temporal landmarks showing when the test stimuli were presented. As specified above, the target heartrate was defined as 80% of maximal heart rate, given the chronological age [13]. The average target heart rate was 150.9 ± 1.0 bpm. Figure 2 shows that for all the three experiments, the heart rate steadily increased during the first 3 min of warm up and then remained constant around the target value for the next 5 min of running.

### 2.7. Data Analysis

Perceptual accuracy (bias) was measured by plotting the proportion of trials in which the test was judged to be the same as the reference, as a function of the test stimulus magnitudes, plotted on a logarithmic axis (examples in Figure 3). These distributions were fitted with Gaussian functions, and the peak of the fitted functions was taken as the “point of subjective equality” (PSE), where the test perceptually matched the reference. This point is the value that the test stimulus had to assume for the subject to have the highest probability to answer “same”. A peak value lower than the physical reference corresponds to the test stimulus being overestimated and vice versa. We describe under- or over-estimations as proportional shifts, defined as the difference between the PSE and the physical value of the reference, normalized by the reference value.
(1)$$\mathrm{Bias}=\left(\frac{\mathrm{Reference}-\mathrm{PSE}}{\mathrm{Reference}}\right)\times 100\%\;$$

We defined perceptual precision as Weber fractions (Wfs), the ratio of the just-noticeable difference (given by the width of the Gaussian fitting function) to the PSE. In practice, the Wfs were computed as the antilog of the standard deviations of the Gaussian log fits minus one.

The data were analyzed with repeated measure ANOVAs, t-tests, Pearson correlations, and bootstrap t-tests [14]. Whenever the sphericity assumption was violated, the Greenhouse–Geisser correction was applied. The standard statistics were complemented with the estimation of Bayes Factors [15], which quantify the evidence for or against the null hypothesis as the ratio of the likelihoods for the experimental and the null hypothesis. We express it as the base10 logarithm of the ratio (Log10Bf10), where negative logarithms indicate that the null hypothesis is likely to be true, positive that it is false. By convention, absolute Log10 Bayes Factors greater than 0.5 are considered substantial evidence for the alternate or null hypothesis, and absolute log factors greater than 1 are strong evidence. All statistical analyses were performed with MatLab 2016b (The Mathworks, Inc., Natick, MA, USA) and Jasp Software (version 0.14.1; JASP Team, Amsterdam, The Netherlands).

## 3. Results

### 3.1. Aggregate Data

As detailed in the methods section, the participants compared a previously viewed reference stimulus lasting 600 ms with a series of test stimuli and judged them as the same or different. In separate sessions, they made similar judgments about the numerosity of the stimuli compared with a 24-dot standard. Figure 3 shows the results of the aggregate data summed over all the participants, as the proportion of “same” responses as a function of test duration or numerosity. The peak of the Gaussian fits describing the distributions reflects the point of subjective equality of test and reference (PSE). A leftward shift of the curve peak compared to the reference value indicates an overestimation of the duration or numerosity of the test stimuli. For all the Gaussian fits on the aggregate subject, an R^2^ higher than 0.97 was achieved.

Figure 3A refers to duration estimates while running. On inspection, it is evident that, compared to the baseline performance before (T1) and after (T2), the visual durations were substantially overestimated during the running phase. While running, a stimulus lasting 513 ms was perceptually judged as equivalent to the 600 ms reference, an overestimation of about 15%. In the two baseline conditions, the peaks were both near the physical reference duration (baseline T1: 608 ms, baseline T2: 588 ms). However, duration perception remained almost veridical when the stimuli were presented soon after the running phase, although heartrate was still elevated well above resting levels (Figure 3B, Test: 556 ms, baseline T1: 572 ms).

Figure 3C reports judgments of numerosity while running compared with the reference of 24 dots. Unlike the duration perception, the numerosity estimations measured while running were almost identical to the baseline conditions (Test: 24.2 dots, Baseline T1: 24.5 dots, Baseline T2: 25.0 dots).

We quantified the significance of the biases of the aggregate data by the bootstrap test. On each repetition (10,000 iterations), and separately for each condition, the data were sampled with the replacement (as many independent samples as the full dataset) and fit with a Gaussian distribution, whose peak yielded an estimate of the PSE. The statistical difference was assessed by comparing the distribution peaks along the bootstrap iterations by Z-test, with the Z-score given by the distance between the distribution means divided by the estimate of the average standard error, given by the square root of the sum of the variances across the bootstraps [14]. Figure 4 shows the distributions of the peak bootstraps for each condition. The average peak for the duration perception while running (Figure 4A, blue distribution) was 513.2 ± 6.7 ms, clearly different from the baseline measured before (T1: 607.9 ± 7.0 ms, Z = 9.7, *p* < 0.0001) and after running (T2: 588.4 ms ±7.6 ms, Z = 7.42, *p* < 0.0001). The two baseline conditions were similar to each other (Z = 1.14, *p* = 0.25). These results confirm that the overestimation of visual duration while running was significantly different from the baseline.

The bootstrap results for the duration estimates after running (Figure 4B) almost overlap those of the baseline and were clearly not statistically different (T1: 571.5 ± 7.7 ms, Test 556.3 ± 7.26 ms, Z = 1.8, *p* = 0.07). For numerosity (Figure 4C), the distributions for the test and the baseline conditions show little to no difference. The two baseline conditions were almost overlapped (T1: 24.5 ± 0.27 dots, T2: 25 ± 0.28 dots, Z = 1.36, *p* = 0.17). The PSE for numerosity while running was slightly lower than the second baseline measured soon after the running phase (Test: 24.2 ± 0.24 dots, T2: 25 ± 0.28 dots, Z = 2, *p* = 0.045 > α = 0.017, Bonferroni corrected for three comparisons). Overall, these results confirm that the running activity distorted duration perception, while numerosity was unaffected.

### 3.2. Individual Data

We also analyzed the data separately for each participant and tested the differences with standard between-participant statistical tests. Using a similar technique to that used for the aggregate data, we fitted each participant’s data with Gaussian functions and estimated the individual PSEs from the peaks in the individual curves. Figure 5 plots, for each participant, the PSE during (or after) running against that of the first baseline.

The points falling below the equality line mean that the PSEs during or after running were lower than the baseline, hence an overestimation. For the duration judgments during running, every single participant falls below the equality line, showing an overestimation of duration. The averages are shown by the arrows and are very similar to the estimates of the aggregate data (overestimation of ~20%). Obviously, this was statistically significant (t(14) = 7.27, *p* < 0.001, Cohen’s d = 1.73, Log10Bf10 = 3.7). The biases in the duration perception after the running phase were scattered much closer to the equality line, with a weak tendency to fall below. However, the effect did not reach significance (t(14) = 1.98, *p* = 0.07, Cohen’s d = 0.54, Log10Bf10 = 0.09). For the numerosity task, there was clearly no tendency for underestimation (or overestimation), with most of the participants scattered around the equality line (t(14) = 0.21, *p* = 0.84, Cohen’s d = −0.019, Log10Bf10 = −0.57). Figure 5D summarizes the individual data, showing the percent biases for the three conditions: averages as bar graphs and individual data as dots. Quite clearly, the only significant effect was duration while running, agreeing with the aggregate data.

While not reaching statistical significance, there was a slight systematic tendency for overestimation of duration after running. One possible explanation is that underestimation occurred reliably soon after the cessation of running but faded quickly. To test this possibility, we separated the baseline and test data (aggregated across the participants) into early and late trials, with a median split of the interval after the running was stopped. The results in Figure 6 show a similar null effect for both halves, suggesting that even in the very first trials after running, the effect was already absent (first half: T1 541 ms ± 9.59 ms, test 528.5 ms ± 9.65 ms, Z = 0.9, *p* = 0.36; second half: T1 604.1 ms ± 11.39, test 585 ms ± 10.22, Z = 1.3, *p* = 0.19).

### 3.3. Precision of Duration and Numerosity Judgments While Running

In order to assess the overall task complexity and difficulty, we analyzed the precision of the participants’ responses, expressed as Weber fractions. Figure 7 shows the Weber fractions (derived from the width of the fitted functions) separately for each participant. ANOVAs and t-tests confirmed that there were no significant differences in the Wfs between the conditions within each experiment (duration while running: F(1.3,18.5) = 2.95, *p* = 0.09; duration after running: t(14) = −0.686, *p* = 0.5; and numerosity: F(2,28) = 0.11, *p* = 0.9). As precision was unaffected by running, we averaged over the baseline and active condition separately for the three experiments. The Weber fractions were clearly higher for the duration conditions than for the numerosity (F(2,28) = 32.39, *p* < 0.001). Post hoc tests revealed similar and not statistically different levels of precision between the “while” and “after” running conditions (t = −0.969, *p* = 0.34), suggesting that this was not the reason for the difference in the results for these two conditions. On the other hand, both duration experiments statistically differed from the numerosity task (both *p*-values < 0.001).

Finally, as much of the evidence suggests common mechanisms for numerosity and duration perception, we looked at the between-task correlations. We computed summary precision indexes separately for numerosity and duration by averaging all the standard deviations across conditions for the two tasks. In line with the involvement of a common mechanism in numerosity and duration perception, the results (Figure 8) showed positive and statistically significant correlations between the precision levels (r = 0.58, *p* = 0.02).

## 4. Discussion

The main result of this study shows that while motor activity can significantly distort the perception of time, it leaves numerosity perception unaffected. The participants compared the duration or numerosity of a test stimulus while running, or having just run, with a standard encoded at rest. The first experiment measuring duration estimation during running supported the previous findings [11], reporting a systematical overestimation of perceived duration while running. The second experiment, however, showed that the duration measurements made soon after stopping running are veridical, showing that this effect is intrinsically related to the movement itself rather than to other physiological parameters altered by physical activity (such as heart rate). In the final experiment, we showed that the numerosity estimation was unaffected by running.

These findings suggest a magnitude-selective interaction between action and perception that involves only the perceived duration. However, one of the most prominent theories on the perception of magnitude posits that the human brain encodes space, time, and number via a shared mechanism [1], opening the possibility that the effect of motor activity on the perception of time might also generalize to other magnitudes. Indeed, it has recently been reported that a repetitive motor routine executed with the upper limbs (hand tapping) distorts both perceived duration [8] and numerosity [9]. Moreover, motor activity related to eye movements has been reported to significantly distort numerosity [16,17].

How can the discrepancy between the present and the previous studies be reconciled? One possibility regards the spatial congruency between the position where the visual stimuli were displayed and the area where the motor activity was directed. The perceived numerosity during saccadic eye movements was distorted only for stimuli displayed between the saccadic starting and ending point, embracing a distance of roughly 20° [16]. Similarly, hand tapping distorted the perceived numerosity of stimuli presented around the tapping area, with effects that rapidly faded off with the increase in the spatial offset relative to the tapping location and completely vanished for distances higher than 15° [8].

Another possible explanation for the null effect found for numerosity may be the different sensory precision for the two magnitudes. The Weber fractions for the numerosity perception were lower than those for the duration perception. It is possible that the noisier system for duration perception is more prone to distortion by contextual variables, such as running. However, we found no difference in the Weber fractions between the duration judgements during and after running, making it unlikely that this is the general explanation for all lack of effects. A last methodological difference worth discussion is the presentation modality of the stimuli. As the dots in the numerosity task were all presented simultaneously, this might have required a lower involvement of the working memory compared to the duration stimuli. However, as other interactions between self-motion and numerosity were reported for both the sequential and the simultaneous [9] numerosities, this difference alone is unlikely to have cancelled out an effect of running on numerosity perception.

Interestingly, in many of the previous studies the participants underestimated perceived duration as a consequence of action. In contrast, the participants in the present study showed a tendency to overestimate the durations of visual stimuli presented during the running phase relative to those with the same physical duration perceived at rest. A possible reconciliation of these discrepancies may be the difference in methodologies. In all the previous experiments, the participants were required to compare two intervals presented one after the other or to immediately reproduce a temporal interval that had been just observed. In the paradigm of the present study, the intervals presented during the running phase were compared with the reference encoded before the onset of physical activity. This resulted in a delay between the encoding (at rest) and the test phase (during running) of at least 3 min, a much longer time for which sensory information had to be stored in the short-term working memory.

A second methodological difference regards the duration and the intensity of the motor activity. While many studies on eye and hand movements use transient motor-routines in the sub-second or supra second regime, here participants were engaged in a strenuous physical activity lasting several minutes. Indeed, a previous report in which participants were required to estimate the duration of visual stimuli during a sustained cycling routine lasting several minutes revealed that the duration estimates during physical exercise were robustly lengthened [11]. Typically, these time-dilation phenomena are accounted for by alterations in the rate of the internal pacemaker: physical activity could accelerate the clock rate, and this, in turn, would induce a perceived dilation of stimulus duration. To reconcile all these results, we might speculate that transient motor activities (such as saccadic eye movements or hand movements) momentarily slow down the rate of the pacemaker, resulting in a compression of perceived time. On the other hand, sustained physical activity may induce an acceleration of the internal clock, yielding the opposite phenomenon of time dilation.

An alternative explanation might be that time processing in different conditions involves several, independent temporal mechanisms in the brain. This idea of multiple clocks in the brain has been demonstrated within the visual domain. A grating drifting at high speed presented in a given location of the visual field strongly compresses the perceived duration of the stimuli subsequently displayed around that area, without affecting those displayed in other locations [18,19]. This finding supports the idea of multiple time mechanisms, each responsible for time processing in a well-defined portion of the visual field, violating the well-held belief of a centralized, unique internal clock. Despite previous reports supporting the idea that motor routines can affect time perception, it remains an open question whether such an interaction is prompted by the execution of the movements themselves or by the alteration of other physiological variables that are perturbated during action.

Past reports have proposed that heart rate might be directly related to the internal clock rate; so, an acceleration of heartrate would also speed up the internal clock, leading to an overestimation of perceived time [20]. To address this possibility, we measured whether time perception was affected not only during the running phase but also immediately after the end of the physical activity. The results indicate that no distortion occurred after completion of the motor routine, even though heartrate had not returned to the baseline level. Even when we took into consideration only the trials immediately after the physical activity, no significant perceived time dilation was observed. This result clearly questions the causal role of heartrate variations in distortions in perceived time and is in line with a previous study reporting significant time dilations when arousal increased but heart rate remained constant, or even decreased [21]. Can the present results be accounted for by variations of the arousal level induced by motor activity? The lack of temporal distortions when duration estimates were made at the end of the physical activity also questions the arousal hypothesis, given that arousal levels have been reported to be still significantly altered at the end of a 10 min running exercise (see [22]) in a similar activity to the present study.

Taken together, the results of the present study reinforce previous studies showing that time perception is affected by running, but a similar running regime does not affect numerosity perception. However, a comparison between these findings with those in the literature revealed that this relationship is modulated by movement parameters, such as movement speed, type of effector, spatial proximity between stimuli, and movement location, as well as the time of stimuli presentation relative to the phases of the motor routine [23]. Future studies should directly investigate each of these issues to provide a full comprehension of the mechanisms and the nature of the interaction between the motor and the sensory systems.

The effect of long-term physical training on time perception should also be investigated. For instance, there have been anecdotical reports from tennis and baseball players that time slows down just before hitting the ball [24]. This suggests that the target of goal-directed actions can benefit from specialized processing of their temporal features, which might be aimed at maximizing motor performance and be induced by the extreme long-term training characterizing elite athletes. However, while the benefit of time dilation for goal-directed action is evident, the same does not hold for target-free rhythmic actions, such as running. This may suggest that while the observed behavioral effect is the same, it might be caused by different mechanisms. During sustained physical exercise, the fatigue accumulated by the participant may cause an overestimation of perceived time, as has also been reported to occur during sustained rowing exercise [25]. In the light of this, future studies could investigate the precision and accuracy of temporal perception in elite athletes, such as those engaged in long-distance races (marathons) or hurdling, combining the running routine with a goal-directed, repetitive, transient action aimed at jumping over the obstacles.

## Figures and Tables

**Figure 1 brainsci-12-00081-f001:**
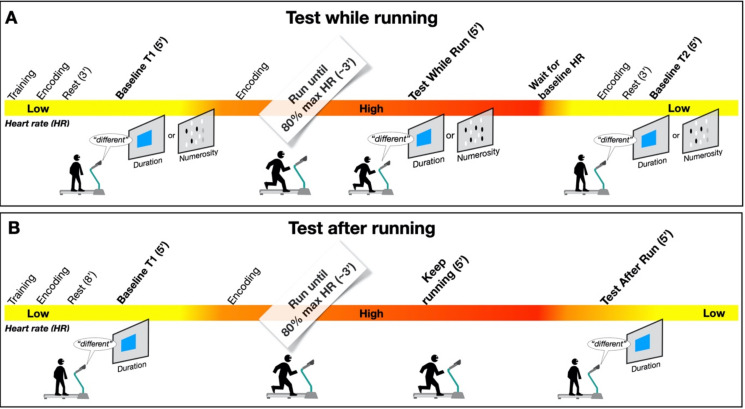
Experimental procedure. (**A**) Paradigm used to measure duration and numerosity perception while running. After a short training session, participants were presented with the reference stimulus (encoding phase: 600 ms or 24 dots) 5 times. After a rest period of 3 min, they were presented with a sequence of test stimuli to categorize as same or different, compared to the reference (baseline T1). After the task, the encoding was repeated, followed by the running phase. During the first three minutes of running, no stimuli were presented; then, the same different task (duration or numerosity) was performed, this time while participants kept running. After this test phase, participants were allowed to rest until the heart rate returned to the baseline level. At this point, a second baseline (T2) was measured. After a short break, the whole procedure was repeated. (**B**) Paradigm used to measure duration perception after running. This was similar to that described above (**A**) except the rest period before baseline measurement (T1) was 8 min instead of 3 min, and the test measurements were made after the participant stopped running while the heartrate reverted to baseline.

**Figure 2 brainsci-12-00081-f002:**
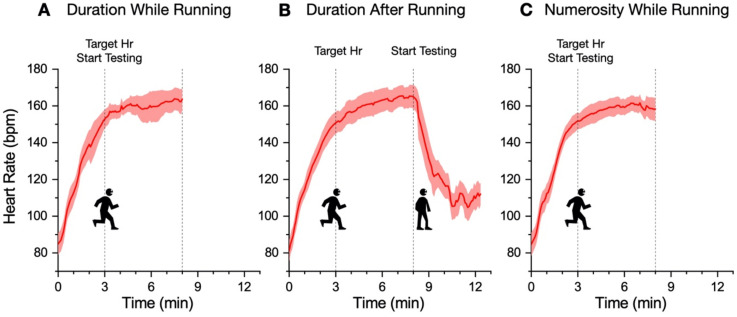
Heartrate parameters. Average heart rate with ±95% C.I. as a function of time after running onset. In all the experiments (**A**–**C**), the heart rate gradually reaches the target value (see methods) within 3 min, then remains stable around that value for the subsequent 5 min of running.

**Figure 3 brainsci-12-00081-f003:**
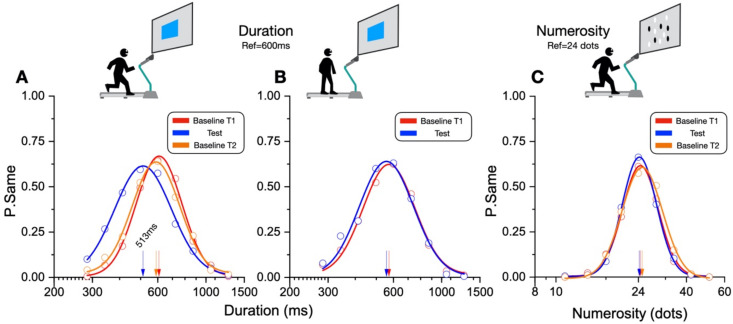
Results on aggregate data for the duration (**A**,**B**) and numerosity (**C**) tasks. Test stimuli magnitudes were plotted against the proportion of “same” responses and fitted with Gaussian functions. The peaks of the fits (arrows) correspond to the PSE (600 ms or 24 dots). A leftward shift (relatively lower peaks values) corresponds to an overestimation of the duration or numerosity of the test stimuli.

**Figure 4 brainsci-12-00081-f004:**
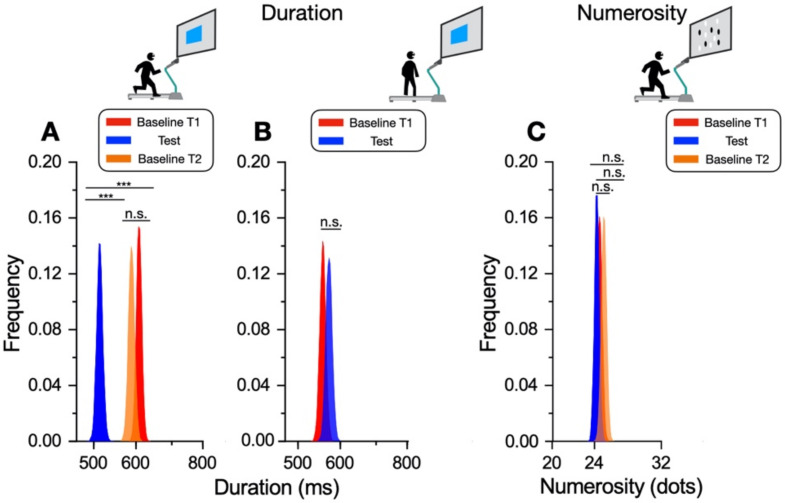
Bootstrap Z-test on aggregate data. Distributions of fitted peaks for the duration (**A**,**B**) and numerosity (**C**) tasks. Overlapped distributions indicate no difference between conditions. *p*-values represent Z-test significance level: *** *p* < 0.01 > α = 0.017, Bonferroni corrected for three comparisons. n.s.–nonsignificant.

**Figure 5 brainsci-12-00081-f005:**
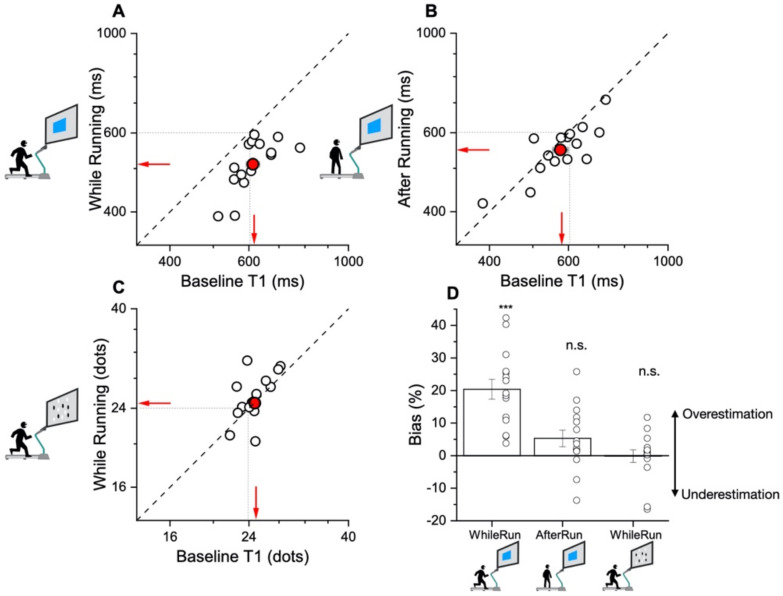
Individual data and perceptual biases magnitudes. Scatter plots showing PSE (log units) separately calculated for each participant (open symbols). Filled symbols refer to group averages. Symbols falling below the equality line (dashed line) reflect lower PSEs in the running condition, hence an overestimation of duration or numerosity. (**A**) Duration PSEs during running against first baseline (T1). (**B**) Duration PSEs after running against first baseline (T1). (**C**) Numerosity PSEs during running against first baseline (T1). (**D**) Bar plot showing estimation biases relative to the baseline condition (T1), normalized by the reference stimulus (600 ms or 24 dots). Bars represent the three experiments: duration while running, duration after running, and numerosity while running, respectively. Bars are between participant’s average, error bars are ±1 SEM. Individual data are represented by symbols. *** *p* < 0.001 n.s. not significant.

**Figure 6 brainsci-12-00081-f006:**
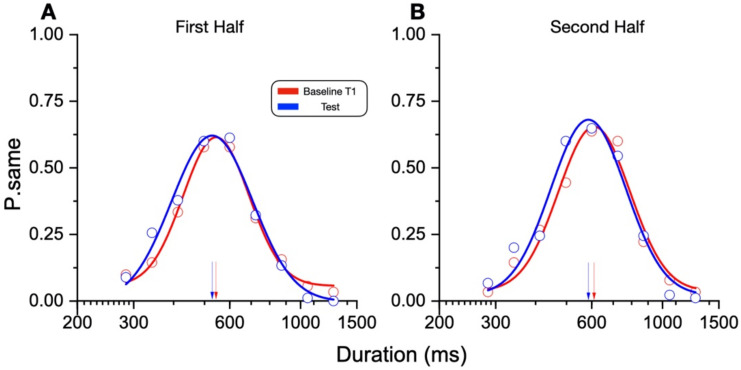
Duration perception after running. As in Figure 3B, stimuli durations were plotted against the proportion of “same” responses and fitted with Gaussian functions with the peak of the fits (arrows) corresponding to the test duration matched with the reference (600 ms). Blue curves report the aggregate data before running (baseline T1), the red curves the data collected after the running phase. To test whether the effect was detectable in the very first trials after the run phase, the analyses were performed on two sub-set of the data: the first (**A**) and second half (**B**).

**Figure 7 brainsci-12-00081-f007:**
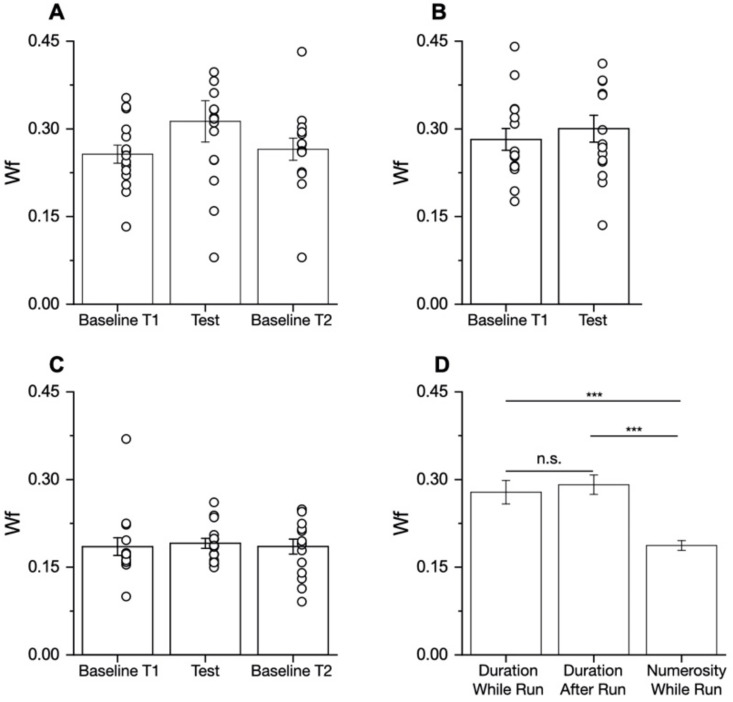
Perception precision. Bar plot showing estimation precision (Wfs) for duration (**A**,**B**) and numerosity (**C**) tasks. Bars show average, error bars are ± 1 SEM, and circles are individual data. (**D**) Average Wfs for the duration and numerosity conditions. *** *p* < 0.001. n.s.–nonsignificant.

**Figure 8 brainsci-12-00081-f008:**
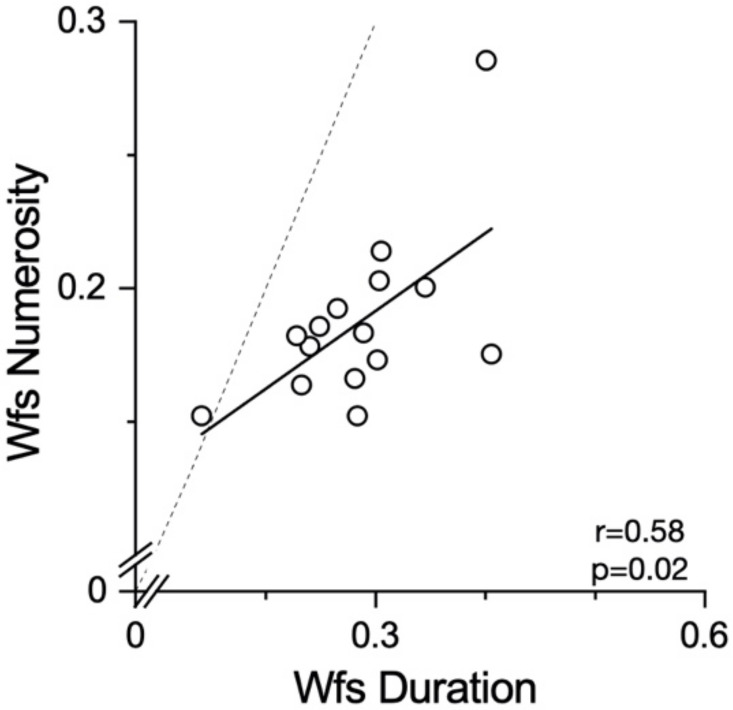
Correlation between numerosity and duration Wfs. Scatter plot of individual Wfs as averaged across conditions for duration and numerosity.

**Table 1 brainsci-12-00081-t001:** Descriptive statistics of running parameters.

Condition	Average HR B/m	Average Speed Km/h	Average Steps	Steps per Second Hz
Baseline (no run)	87.06 ± 2.1	--	--	--
Duration while running	159.6 ± 0.67	7.87 ± 0.66	1185.5 ± 20.8	2.47
Duration after run	160.4 ± 1.1	7.61 ± 0.62	1206.6 ± 24.67	2.51
Numerosity while running	158.1 ± 0.31	7.65 ± 0.63	1217.3 ± 27.8	2.53

## Data Availability

Data for the main findings are available at: https://doi.org/10.5281/zenodo.5796552 (accessed on 30 November 2021).

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
