# Peer review of "Visual Duration but Not Numerosity Is Distorted While Running"

_brainsci, 2022, doi:10.3390/brainsci12010081_

Round 1
Reviewer 1 Report
The study investigates how activity influences the perception and processing of magnitudes such as duration and numerosity. As the effect of exercise on numerosity perception has not yet been investigated, it is a central part of the study. Participants were asked to make a temporal or numerical judgement, at rest and while running. Results show an expansion of perceived duration with physical activity, while there was no such effect on perceived numerosity.
Although I believe that the topic is very interesting, the manuscript needs major revisions before publishing. There are especially concerns and uncertainties regarding described methods. See my detailed comments below.
Abstract
- Please re-check the abstract for given information. Line 16 mentions body temperature as a physiological factor, however, it is not investigated in the study.
Introduction
- While the introduction gives important information on the topic, it needs more streamlining. Why are some of the studies from the discussion not mentioned before. Studies like Lambourne (2012) should already be cited and explained in the introduction to give a better insight into already existing research in the field. The hypotheses at the end of the introduction part (ll.66-74) should the phrased more clearly. From what prior research is what aspect of the hypotheses concluded? What is already known and what is new in the current study?
- Could the authors provide a paragraph in the Introduction part on why the effect of exercise on numerical judgement is relevant? What are practical implications? As this is the only new research, there should be a stronger focus on this part.
Methods and results
I see major concerns with the described methods and think that vital information is missing.
- Can you please explain what the additive “four authors” (l.78) means in this context?
- How were ages in the sample distributed. Only a mean but no standard deviation is given. I would kindly ask the authors to add the SD. I am afraid that considerable age gaps would significantly influence data analysis and interpretation.
- The sample size is very small and leads to lower power of the data. The authors should specify why they chose this sample. I am missing power and sample size estimation. Especially, since null effects/ beta-errors are interpreted in the ANOVAs.
- From line 100 to 107 a training phase is described, where all participants had to reach 80% accuracy before proceeding to the encoding phase. Could the authors provide data on how many training runs each participants needed. Was this matched to later estimations. I would think that one who has trouble with estimation in the training run would also rather make more mistakes in later perception.
- Why were 600ms as a reference for duration and 24 dots used as a reference for numerosity? Would the results change with a different reference?
- How were durations of rest and test periods chosen?
- Why were logarithmic distances between the test stimuli chosen?
- What is the difference between test after run and baseline T2?
- The heart rate was measured so accurately, was there a correlation between the heart rate and the overestimation of the duration? I find no direct link between the two in the manuscript. It would be important to know, if the level of fitness directly influences perception – which I think it does.
- Could the authors please give further explanation for the “point of subjective quality”? This aspect is unclear. The major concern I see with the methods section is where data comes from. How were responses measured? Did the participants really only decide whether the test stimuli differed from the reference by “same” or “different” or were they asked for their exact judgement of duration in ms or number of dots given on the screen. I would see a problem with exact numerosity judgement, as stimuli were only given for 200ms. I see that authors wanted to prevent the participants from counting. But one would not be able to capture 51 dots at once (see for example Svenson & Sjoberg, 1983 on subitizing effects). If judgements were really just “same” or “different”, how was the data generated that was used for the statistical analyses.
- Following up with comment 6 – could the authors please give effect sizes for all statistical analyses.
Discussion
- The discussion is actually very detailed and well-structured. Yet, the mentioned references should be matched between introduction and discussion.
Additional comments
- We strongly recommend that authors check the manuscript for wording and grammatical aspects such as sentence order, as there are numerous errors here that hinder the flow of reading. (see for example lines 28/29, 33/34)
Reviewer 2 Report
I very much enjoyed reading this excellent paper. It is well written, well conducted and thoroughly discussed. I have a few comments for the authors, critically one regarding the presentation mode of duration vs. number.
In the Introduction, the numerical distance effect is given as an example for the interactions between different magnitude dimensions; however, how is this effect related to other magnitudes? The distance effect does not provide evidence for an interaction between number processing and other magnitudes, but tells us about the functional properties of numbers. If this example is aimed at exemplifying the interaction between number and space, there are other phenomena/studies that should be cited instead (e.g., the SCE).
Typo: Participants: ‘A total for’ should read ‘A total of’.
How was sample size determined? How did the authors make sure that this sample (15 participants) was large enough to detect an effect of numerosity? Processing duration and number differs in many aspects; therefore it is not enough to find an effect in one dimension to infer that the sample was appropriate to detect the effect in another dimension. Moreover, in this experiment numerosity was not presented sequentially but all dots appeared at once. Therefore, memory was playing a more relevant role in the duration experiment than in the number experiment. This variable alone, the presentation modality, could explain why duration was distorted but not numerical processing. The authors might want to add this methodological aspect to their discussion.
